# Terpenes modulate bacterial and fungal growth and sorghum rhizobiome communities

Ming-Yi Chou,[1,2,3] Trine B. Andersen,[2,4] Marco E. Mechan Llontop,[2,5] Nick Beculheimer,[2,5] Alassane Sow,[5] Nick Moreno,[4] Ashley Shade,[1,2,5,6] Bjoern Hamberger,[2,4] Gregory Bonito[1,2,5]

**ABSTRACT** Terpenes are among the oldest and largest class of plant-specialized bioproducts that are known to affect plant development, adaptation, and biological interactions. While their biosynthesis, evolution, and function in aboveground interactions with insects and individual microbial species are well studied, how different terpenes impact plant microbiomes belowground is much less understood. Here we designed an experiment to assess how belowground exogenous applications of monoterpenes (1,8-cineole and linalool) and a sesquiterpene (nerolidol) delivered through an artificial root system impacted its belowground bacterial and fungal microbiome. We found that the terpene applications had significant and variable impacts on bacterial and fungal communities, depending on terpene class and concentration; however, these impacts were localized to the artificial root system and the fungal rhizosphere. We complemented this experiment with pure culture bioassays on responsive bacteria and fungi isolated from the sorghum rhizobiome. Overall, higher concentrations (200 µM) of nerolidol were inhibitory to *Ferrovibrium* and tested Firmicutes. While fungal isolates of *Penicillium* and *Periconia* were also more inhibited by higher concentrations (200 µM) of nerolidol, *Clonostachys* was enhanced at this higher level and together with *Humicola* was inhibited by the lower concentration tested (100 µM). On the other hand, 1,8-cineole had an inhibitory effect on *Orbilia* at both tested concentrations but had a promotive effect at 100 µM on *Penicillium* and *Periconia*. Similarly, linalool at 100 µM had significant growth promotion in *Mortierella*, but an inhibitory effect for *Orbilia*. Together, these results highlight the variable direct effects of terpenes on single microbial isolates and demonstrate the complexity of microbe-terpene interactions in the rhizobiome.

**IMPORTANCE** Terpenes represent one of the largest and oldest classes of plant-specialized metabolism, but their role in the belowground microbiome is poorly understood. Here, we used a "rhizobox" mesocosm experimental set-up to supply different concentrations and classes of terpenes into the soil compartment with growing sorghum for 1 month to assess how these terpenes affect sorghum bacterial and fungal rhizobiome communities. Changes in bacterial and fungal communities between treatments belowground were characterized, followed by bioassays screening on bacterial and fungal isolates from the sorghum rhizosphere against terpenes to validate direct microbial responses. We found that microbial growth stimulatory and inhibitory effects were localized, terpene specific, dose dependent, and transient in time. This work paves the way for engineering terpene metabolisms in plant microbiomes for improved sustainable agriculture and bioenergy crop production.

**KEYWORDS** monoterpene, sesquiterpene, amplicon sequencing, bacterial 16S rDNA, fungal ITS rDNA, root microbiome

Address correspondence to Gregory Bonito, bonito@msu.edu.

Ming-Yi Chou and Trine B. Andersen contributed equally to this article. Order was determined by drawing a name out of an opaque bag.

The authors declare no conflict of interest.

See the funding table on p. 18.

Terpenes are recognized as the largest and oldest class of plant-specialized bioproducts and are important for plant development, adaptation, and interaction with the environment. Their functions, biosynthesis, and evolution in aboveground interactions with individual insects and individual microbial species have been reported across the plant kingdom (1–4). Terpenoids are divided into subclasses depending on the number of carbon atoms in their backbones: monoterpenoids (C10), sesquiterpenoids (C15), diterpenoids (C20), sesterterpenoids (C25), triterpenoids (C30), and carotenoids (C40). How these different terpene subclasses impact plants and their microbiomes is far more complex and less understood, despite wide-ranging ecological and plant fitness implications.

Recent studies have demonstrated that terpenes modulate the composition of the floral microbiome and that the microbiome itself can contribute to or induce floral terpene emissions (5, 6). The first evidence for terpene-based modulation of the root microbiome emerged in 2019 when triterpene (C30)-specialized metabolites were found to control the composition of the *Arabidopsis thaliana* bacterial root microbiome (7). Just 2 months later, the discovery of root-specific sesterterpenes (C25) was reported to selectively regulate the assembly of the terpene-sensitive root microbiome, again in *A. thaliana* (8). Terpenes can also play roles in fungal-bacterial interactions (9), for example, the fungus *Fusarium culmorum* can produce specialized terpenes that affect the motility of some common soil bacteria (10) and a terpene synthase from *Serendipita indica* yielded a product with antifungal activity against a phytopathogen (11). Thus, there are enormous biotechnological opportunities for plant protection and boosting productivity through the rational design of synthetic microbial communities. In the aboveground tissues of sorghum, a limited complexity of triterpenes was reported (12), with a proposed role in the protection of water loss at high temperatures. Similarly, in leaves of sorghum, the sesquiterpene metabolism responds to insect herbivory by increased expression of genes encoding the respective terpene synthases and accumulation of their products (13). While the terpene metabolism in the root of switchgrass and maize has been well studied, with suggested roles in plant defense or assembly of the microbiome (14, 15), the corresponding metabolism in the sorghum root is currently not understood.

Sorghum (*Sorghum bicolor*) has attracted attention as a biofuel crop given its vigorous growth, resilience, and genetic tractability. Furthermore, being an annual crop bioenergy sorghum may offer flexibility to farmers who want to transition into bioenergy crops. Bioenergy sorghum has deep roots (up to 2 m) that continue growing and taking up water during long periods of water deficit, and even when the top 50 cm of the soil profile is dry. Interestingly, small droplets on sorghum root hairs, colonized by microbes, were reported in 1978 (16). It was subsequently shown that the droplets consist nearly exclusively of the hydrophobic benzoquinone sorgoleone and derivatives, suggested to be synthesized at the endoplasmic reticulum (ER) before exudation (17).

In this study, we ask how three commercially available terpenes impact plant-associated fungi and bacteria, as individuals and in a community setting. To address this experimentally, we deployed a simulated root exudation system, engineered from micro-rhizon sampling devices (Rhizosphere Research Products, Netherlands) (18), in mesocosm growth chambers where sorghum was grown in field soil from sorghum plots (Fig. S1). This system allowed us to augment the rhizosphere with the three different terpenes chosen based on the four criteria of (i) their formation in plants, (ii) known biosynthesis, (iii) commercial availability, and (iv) water solubility (Table 1) (5, 19–30). Total treatment required 3–6 mg of pure standardized terpenoid. Of the three terpenoids, one was the cyclic monoterpenoid alcohol 1,8-cineole, while the other two were the linear alcohols: the monoterpenoid linalool and the sesquiterpenoid nerolidol (Table 1). To determine specific responses to the supplemented terpenes, both fungal and bacterial communities of the bulk soil, root endophytes and the rhizosphere, were characterized through community amplicon sequencing. To further investigate the individual responses of fungal and bacterial species, the growth of identified pure

**TABLE 1** Cyclic (1,8-cineole) and acyclic (nerolidol and linalool) target terpenes and known activities

| Target | Biosynthetic pathway (plant/microbe) | Evidence for activity |
|---|---|---|
| 1,8-Cineole (monoterpenoid) | Cineole synthases, *Salvia officinalis* (common sage) (20)/*Streptomyces clavuligerus* (21) | Root-specific release in pathogen response in *A. thaliana*; antimicrobial, allelopathic (19) |
| 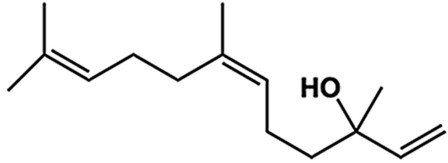 | | |
| Nerolidol (sesquiterpenoid) | Nerolidol synthases, Snapdragon (*Antirrhinum majus*) (25)/*Trichoderma harzianum* (22) | Induced by root bacteria in grapevine (*Vitis vinifera*); drought protectant; active against bacterial and fungal phytopathogens (23, 24) |
| 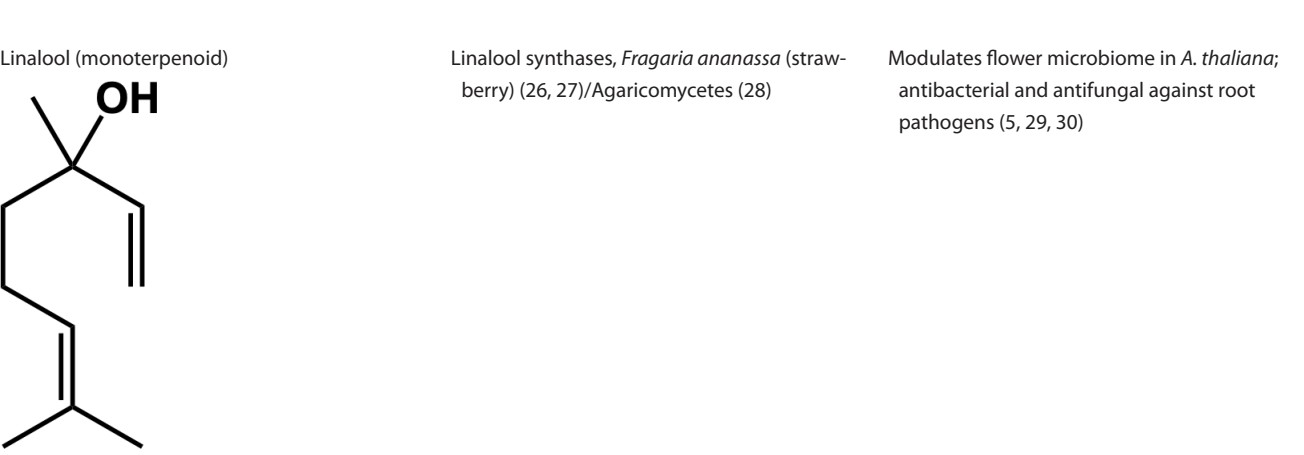 | | |
| Linalool (monoterpenoid) | Linalool synthases, *Fragaria ananassa* (strawberry) (26, 27)/Agaricomycetes (28) | Modulates flower microbiome in *A. thaliana*; antibacterial and antifungal against root pathogens (5, 29, 30) |

isolates of root-associated fungi and bacteria from sorghum was screened against these same terpenes. We hypothesized that due to the different bioactivities of the terpenoids, each terpene class would shift the root- and soil-associated bacterial and fungal community structures differently. By understanding the responses of individuals and communities of diverse fungal and bacterial taxa to these terpenoids, this study sets the stage for future bioengineering of plant terpenoids production to improve beneficial microbial recruitment, pathogen repellence, and ultimately enhanced crop production (31).

## MATERIALS AND METHODS

### Chemicals

Linalool, 1,8-cineole, and cis/trans-nerolidol were purchased from Thermo Fisher Scientific (Thermo Fisher, Waltham, MA, USA).

### Plants growth

Sorghum seeds (Sorghum bioenergy variant Tx08001, William Rooney, Texas A&M, USA) were stratified for 2 days and germinated on wet tissue paper after which the seedlings were transferred to pots and grown for a week to achieve two true leaves. Fresh field soil used in this study was obtained from the top 15 cm profile of sorghum plots that are part of the Bioenergy Cropping Systems Experiment at the Kellogg Biological Field Station, Hickory Corners, MI, USA. Young plants of the same size were selected and transferred to rhizoboxes, tri-compartmented acrylic boxes (54 cm × 4 cm × 30 cm) with transparent covers allowing the visualization of root growth, filled with sorghum field soil mixed with double autoclaved play sand (50/50 by volume). Each rhizobox had three compartments with plastic dividers between compartments. Four terpenoid treatments, 1,8-cineole 100 µM, linalool 100 µM, linalool 200 µM, and cis/trans-nerolidol 100 µM, were used in this study along with a box where only water was added as a control, and a control box with no plants also with only water added. The three terpenoids were diluted with water to obtain the selected concentrations. For nerolidol, the 100 µM was delivered in a solution containing well-homogenized dissolved terpene, and undissolved oil as the targeted concentration is beyond the nerolidol water solubility. A recent study demonstrated the direct delivery and diffusion of an insoluble terpenoid in soils of different moisture levels and demonstrated bioactivity (32). The plants were grown in a growth chamber with 25°C/18°C day/night and 16/8-h light/dark cycles. One month after the seedling transfer, terpenoid treatment commenced. Each terpenoid treatment was applied to one rhizobox through the artificial root system (Rhizosphere Research Products, Netherlands) at a rate of 10 mL every other day. This artificial root system simulated the change of terpenoid concentration in a localized condition, similar to terpenoids produced in native root systems. During alternative days of the treatment applications, irrigation was done according to a standard soil humidity test with a soil moisture probe. Sorghum bulk soil, rhizosphere soil, and root were sampled 1 month after seedlings transplant prior to terpene application, and 2 months post-onset of the terpene treatments.

Terpenoid impacts on plant growth were also measured after the final sampling of the microbial community. Growth parameters include a number of healthy, unhealthy, and dead leaves, height, and biomass of aerial and belowground tissues. The sorghum plants were harvested and washed with tap water to rinse off the soil particles. Aboveground and belowground tissues were separated, wrapped in paper bags, and dried in an oven at 70°C for a week. Fresh and dry biomass of aboveground and belowground tissues was measured with a balance. Statistical analyses were conducted in JMP Pro14 (SAS Institute, Cary, NC, USA) and applied to the collected growth parameters using one-way ANOVA followed by Dunnett's test, and Wilcoxon test followed by Dunn's test to better account for small sampling size.

### Microbiome sampling

Sorghum bulk soil, rhizosphere soil, and root were sampled 1 month after seedlings transplant prior to terpene application, and 2 months post-onset of the terpene treatments. The following sampling procedure was deployed at each sampling occasion. Bulk soil was collected with a sterile spatula avoiding areas close to roots. Rhizosphere soil was sampled by randomly collecting three young lateral roots (including the root hairs) up to 3 cm, aggressively agitating them to detach loosely attached soil, and then washing and vortexing the roots in ddH$_2$O containing 0.05% Tween 20 for 20 min. The wash liquid was then collected as rhizosphere soil. Roots from the rhizosphere soil wash

were kept and cleaned with 6% hydrogen peroxide solution for 30 s and rinsed with ddH$_2$O three times. These cleansed surface-sterilized roots were sampled as endosphere communities. The artificial rhizosphere sample was collected by swabbing the surface of the artificial roots after aggressively agitating the artificial roots to remove the visible soil aggregates. The liquid retained from the swabs that were cut was placed into 2 mL tubes containing a 0.05% Tween 20 solution, and vortexed for 20 min. Collected samples were flash frozen in liquid N upon collecting, dried with SpeedVac (Thermo Fisher, Waltham, MA, USA) while freezing, and bead-beaten with TissueLyser II (Qiagen, Hilden, Germany) at maximum speed for 40 s. Microbial DNA was extracted with MagAttract PowerSoil DNA Kit (Qiagen, Hilden, Germany) and E.Z.N.A. Plant DNA Kit (Omega Bio-Tek, Norcross, GA, USA) for soil and plant samples, respectively.

The libraries were prepared according to Beschoren da Costa et al. (33) with some modifications. Briefly, extracted sample DNAs were amplified with primer sets 515f, 806r, and 5.8f, ITS4r for bacterial and fungal communities, respectively (34, 35). The amplicons were then attached to sequencing adapters and customized barcodes. The amplicons were then normalized with Norgen DNA Purification Kits (Norgen Biotek Corp., Thorold, ON, Canada), pooled, concentrated with Amicon centrifugal units (Sigma-Aldrich, St. Louis, MO, USA), and purified with HighPrep PCR Clean-up System (MAGBIO Genomics, Gaithersburg, MD, USA). The libraries were then submitted to Michigan State University Genomic Cores (East Lansing, MI, USA) to sequence on Illumina MiSeq v3 kit. The raw sequences were demultiplexed, filtered, clustered into operational taxonomic units (OTUs) at 98% similarity, and taxonomically assigned according to Beschoren da Costa et al. (33). The initial taxonomic ranks were assigned in USEARCH v11 (36) with SILVA v138 (37) and UNITE 9.0 (38), and the final taxonomic ranks were verified with constax2 v2.0.18 (39). Visual and statistical analyses were performed in R 4.1.0 with packages vegan 2.5.7, phyloseq 1.38.0, and ggplot2 3.3.3. The differential relative abundance analyses were done in the software Statistical Analysis of Taxonomic and Functional Profiles v2.1.3 (40) with Welch's $t$-test on OTUs with at least 0.1% by comparing each treatment with water control without multiple comparison adjustments on the $P$-values due to the exploratory nature of this study.The DNA sequences of isolated microbes were blasted against the OTUs derived from the above-mentioned procedure using BLASTn on NCBI.

## GC-MS analysis

Bulk soil from around the artificial root was sampled at the same time as microbiome samples. This soil was extracted O/N with ethyl acetate and analyzed on an Agilent 7890A GC with an Agilent VF-5ms column (30 m × 250 µm × 0.25 µm, with 10 m EZ-Guard) and an Agilent 5975C detector. The injection port was set to 250℃ with splitless injection, and helium was used as a carrier gas with a column flow of 1 mL/min. The oven program was set to 40℃ for 1 min then increased to 90℃ at a rate of 30℃/min, then 90℃ to 110℃, at a rate of 5℃/min, then 110℃ to 165℃ ata rate of 40℃/min, then 165℃ to 180℃ at a rate of 5℃/min and finally 180℃ to 320℃ at a rate of 40℃/min and a 2 min hold. The detector was activated after a 4-min solvent delay. The ionization electron energy was 70 eV. The ion source temperature was 230℃, with an interface temperature of 280℃.

## Fungal isolation and identification

After 1 month of terpene application, sorghum roots up to 5 cm from the root tip were removed, agitated to detach the loosely attached soil particles, placed into 15 mL tubes containing ddH$_2$O with 0.05% Tween 20, and vortexed for 20 min to remove any remaining soil particles. The sorghum roots were rinsed with sterile ddH$_2$O, shaken for 1.5 min in a 3% hydrogen peroxide solution for 30 s, and rinsed again three times with sterile double-deionized water. To isolate endophytic fungi from these surface-sterilized roots, fine root hairs were axenically picked using fine-tip forceps and immersed into malt extract agarose plates containing the three antibiotics chloramphenicol, streptomy-cin, and rifampicin (MEA+). To isolate fungi associated with the artificial roots, these

roots were individually swabbed with sterile cotton swabs and placed into 15 mL tubes containing a tween 20 solution to disperse the particles. The tween 20 solution was then vortexed for 10 min; and 150 µL of the solution, the sorghum root tween 20 solution, and the artificial root liquid were individually pipetted and spread onto separate MEA + plates. To selectively isolate *Orbilia*, a nematode-trapping fungus that was observed to be responsive to some terpene treatments, we sprinkled soil from rhizoboxes onto water agar plates with triple antibiotics and then baited them with surface-sterilized mycophagous nematodes (*Aphelenchus avenae*) that had been reared on *Fusarium* sp. Plates were checked daily under a compound microscope and once conidia were observed a fine needle was used to transfer a single bundle of conidia to MEA + plates. Fungal colonies that grew on the MEA + plates from real and artificial roots, and nematodes, were subcultured individually onto new plates of malt extract agar without antibiotics (MEA) until isolates were pure. All isolates were preserved in a 30% glycerol solution at −80°C, an MEA slant at 4°C, and a scintillation vial containing water at room temperature.

DNA from each isolate was extracted by placing a small amount of fungal tissue in an alkaline-Tris solution and incubating at 95°C for 10 min to lyse the cells (41). The supernatant was used for PCR amplification with the primers ITS1f and LR3 (42, 43). PCR products cleaned with Exonuclease and Antarctic Phosphatase enzymes (Sigma-Aldrich, St. Louis, MO, USA) were sent to Michigan State University Research Technology Support Facility Genomics Core (East Lansing, MI, USA) for Sanger sequencing. Sequence results were trimmed of low-quality reads and aligned into consensus when possible with Geneious 2021.2.2, then compared to pre-existing genomes in the NCBI nucleotide collection for identification.

## Bacterial isolation and identification

Sorghum roots were collected as described above. The tween 20 solutions were serially diluted up to $10^4$ times. One hundred microliters of each dilution was plated on Reasoner's 2A (R2A: yeast extract 0.5 $gL^{-1}$, proteose peptone No. 3 0.5 $gL^{-1}$, casamino acids 0.5 $gL^{-1}$, glucose 0.5 $gL^{-1}$, soluble starch 0.5 $gL^{-1}$, sodium pyruvate 0.3 $gL^{-1}$, $K_2HPO_4$ 0.3 $gL^{-1}$, $MgSO_4 \times 7H_2O$ 0.05 $gL^{-1}$, and agar 15 $gL^{-1}$) agar plates and incubated at 25°C and 37°C for up to 10 days. Individual colonies were transferred onto new R2A plates. Glycerol stocks (25% vol/vol) of pure bacteria isolates were stored at −80°C for future assays. To extract the bacterial genomic DNA, pure bacterial cultures were grown in ½ dilution in water of Tryptic Soy Broth (50TSA: casein peptone 15 $gL^{-1}$, soy peptone 5 $gL^{-1}$, sodium chloride 5 $gL^{-1}$, pH 7.3) at 28°C for 24 h. The bacterial culture was centrifuged at 5,000 rpm for 5 min and genomic DNA was extracted using a Extract-N Amp protocol. We performed PCR amplification of the V4 region of the 16S rDNA gene with universal primers 515F (5′-GTGCCAGCMGCCGCGGTAA- 3′) and 806R (5′-GGACTACHVGGGTWTC-TAAT-3′) using the Pfu Turbo DNA polymerase (Agilent Technologies, Santa Clara, CA, USA) under the following conditions: 95°C for 3 min, followed by 30 cycles of 95°C for 45 s, 50°C for 60 s, and 72°C for 90 s, with a final extension at 72°C for 10 min. PCR products were purified with ExoSAP IT reagent (Thermo Fisher, Waltham, MA, USA), and Sanger sequencing was completed by the Genomics Core of the Research Technology Support Facility at Michigan State University (East Lansing, MI, USA).

High-quality forward and reverse sequences were aligned and trimmed to generate a consensus sequence using Geneious 2021.2.2 (https://www.geneious.com/). Isolates were identified by comparing consensus sequences against the NCBI database through BLASTn searches.

## Terpene screening by selected fungi and bacteria

We screened for terpene impacts on the growth of isolated bacteria and fungi by subculturing selected isolates on the terpene growth media amended with different terpenes as described below. The selection criteria were as follows. Fungal and bacterial isolates had a 100% match with one or more OTU sequences from amplicon sequence analysis of the rhizobox experiment were selected. Selected isolates were represented by

>0.1% in relative abundance in the community and a difference in relative abundances between treatment and water control. Biocontrol and plant-growth promoter fungal isolates were prioritized including *Clonostachys* sp. GLBRC1161, *Fusarium* sp. GLBRC1136, *Humicola* sp. GLBRC1185, *Mortierella alpina* GLBRC1143, *Orbilia conoides* GLBRC1199, *Penicillium* sp. GLBRC1197, *Periconia* sp. GLBRC1140, *Serendipita* sp. GLBRC1134, and *Trichoderma* sp. GLBRC1156. Bacterial strains were prioritized through a non-replicated pre-screen assay.

To determine the growth of bacterial strains in response to the presence of terpenoids, we measured bacterial growth rates over time. In summary, bacterial isolates were streaked out from −80°C on R2A plates and incubated at 28°C for 24 h. Single colonies were grown in 50TSB at 28°C for 24 h and the resulting bacterial culture was pelleted at 10,000 $g$ for 5 min. Bacteria were resuspended in 10 mM $MgSO_4$ and diluted to optical density ($OD_{600}$) = 0.7. Bacteria were inoculated in 50TSB supplemented with either linalool, 1,8-cineole, or nerolidol to a final concentration of 200 μM in a 96-well assay plate. Two controls were used: the first was an uninoculated liquid growth medium supplemented with terpenoids only, and the second was a liquid growth medium with bacteria but no terpenes. For a preliminary growth rate screen, bacterial isolates were grown in a single-replicate assay per condition. Bacterial isolates that responded to the presence of terpenoids in the pre-screening were included in a subsequent three-replicate assay with terpenoids at the final concentrations of 100 μM and 200 μM. The 96-well plate was then shaken in a thermostated Tecan SUNRISE microplate reader (Tecan Austria GmbH, Austria) at 30°C for about 48 h in total. Setting conditions were assigned by the Magellan software (Tecan Austria GmbH, Austria) and the raw OD data were exported as tables with the addition of kinetic time and temperature values. Raw data were imported into R version 4.1.3. Bacterial growth curves were generated using a modified script (https://github.com/briandconnelly/growthcurve).

The growth media used for fungi was 0.1X potato dextrose agar (PDA) for fungi and R2 for bacteria. The terpenes were pre-dissolved in sterile H2O, sterile filtered with 0.2 μm pore size, and added to the 0.1X PDA after autoclaving and cooled down to approximately 50°C prior to pouring into 5 cm radius Petri dishes. The media contained one of the terpenes: 1,8-cineole, linalool, and nerolidol at 100 μM and 200 μM dosages to compare with no-terpene controls. The growth media were inoculated by the placement of a 3 mm radius plug from the edge of 5- to 7-day-old subcultured isolates and wrapped with parafilm to prevent drying out while they grew on the amended media. Colony growth diameters of the fungi were measured twice per dish with the ruler line running through the center of the inoculation plug and with the second measurement perpendicular to the first. Colony diameters were measured every 24 h at 1, 2, and 4 days or until they reached the edge of the dish, whichever occurred first. The diameter of the plug was subtracted from the colony diameter before normalizing to the mean of the control. The significance of these normalized values for fungi was determined by comparing each treatment with the control using the student's $t$-test at α = 0.05 and visualized in R 4.1.0 with package ggplot2 3.3.3.

## RESULTS

### Sequence data

In this study, we generated 16S V4 and ITS2 rDNA amplicon-sequencing data from a total of 36 bulk soil samples, 30 sorghum rhizosphere samples, 18 artificial rhizosphere soil samples, and 30 sorghum root endosphere samples, which yielded a total of 5,740,775 and 8,873,219 reads, respectively, for the 16S and the ITS rDNA. After quality filtering, each sample was left with an average of 49,066 reads for 16S and 73,332 reads for ITS sequences. Overall, we detected 10,988 bacterial and 2,267 fungal OTUs.

## Bacterial and fungal diversity and measurement of plant parameters

Across all OTUs, there were 25 bacterial and 18 fungal classes that represented at least 1% of the overall relative abundance in sorghum rhizobiome. The most abundant bacterial classes in the sorghum rhizobiome include Gammaproteobacteria, Alphaproteobacteria, Actinobacteria, Bacteroidia, and Thermoleophilia (Fig. S2). There were clear differences between root and soil samples. For example, in sorghum roots, Gammaproteobacteria were the most dominant class, while Acidimicrobiia and Acidobacteriae were absent. For fungi, Sordariomycetes were the most abundant class in all sample types but clear differences in composition were evident between sample sources. In the roots, the relative abundance of Agaricomycetes and Glomeromycetes was enriched (Fig. S3). There was also enrichment in Pezizomycetes in bulk soil and Glomeromycetes in the artificial rhizosphere samples where nerolidol was injected. The microbial community structure of post-treatment samples was not clearly separated and did not cluster by treatment for any of the compartments on two-dimensional principal coordinates analysis (Fig. 1). However, the permutational multivariate analysis of variance (PERMANOVA) did indicate that terpenoid application affected the fungal and bacterial communities in the artificial rhizosphere ($P = 0.039$ for fungi and $P < 0.0005$ for bacteria), and fungal communities in the rhizosphere ($P < 0.0005$) (Table 2). Despite variation in read number, the compartments that had significant treatment effect explained more variance than the read number (Table S1). Together, these results indicate treatment effects. We did not detect any impact on the bacterial or fungal community structure of bulk soil and root endosphere by any of the terpenoids applications.

For plant growth parameters, no consistent impacts were observed across experimental treatments (Table S2). However, a greater number of healthy leaves were observed in 200 µM linalool treatments, and slightly greater belowground dry biomass was found in 100 µM nerolidol amendment compared with the water control, respectively.

## Impact of terpenes on specific taxa of bacteria and fungi in a community setting

Given the treatment effect of terpenes on the artificial rhizosphere compartment, we assessed community shifts in further detail. At the OTU level, numerous bacterial and fungal taxa were found to be affected by the application of various terpenes, resulting in

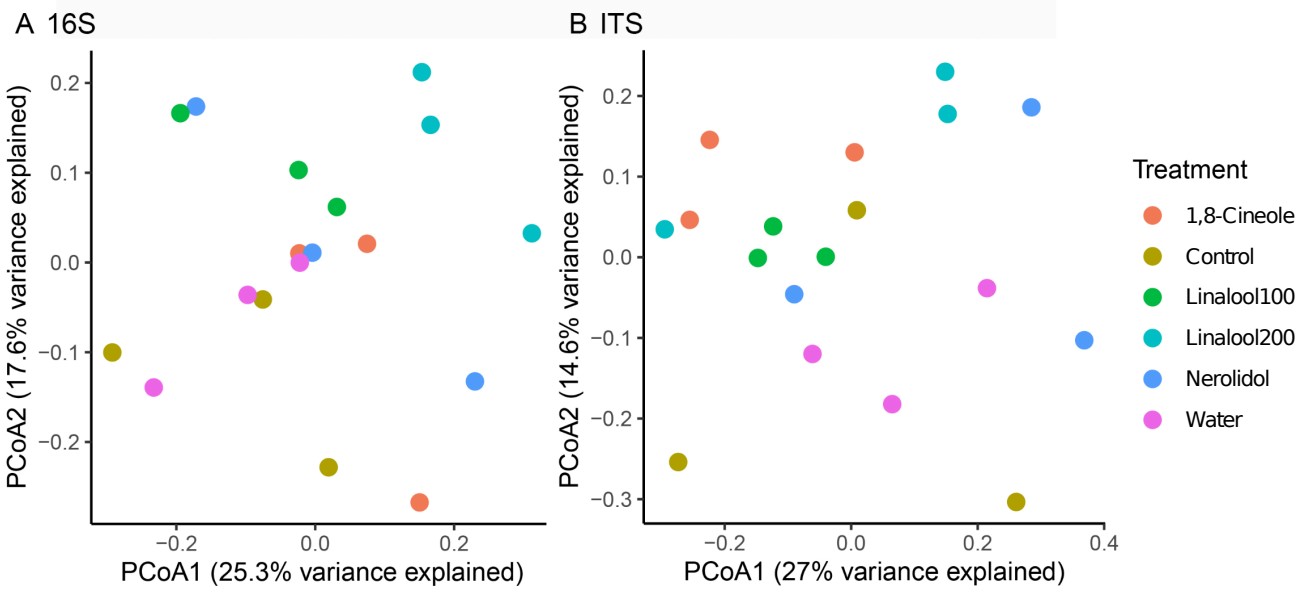

**FIG 1** Principal coordinate analysis of sorghum artificial rhizosphere (A) bacterial 16S and (B) fungal ITS community Bray-Curtis distance after 1 month of application of terpenoids: 1,8-cineole at 100 µM, linalool at 100 and 200 µM, and nerolidol at 100 µM along with water control.

**TABLE 2** PERMANOVA test of the terpene amendment effects on the microbiome of each sorghum belowground compartment

| | 16S | | ITS | |
|---|---|---|---|---|
| | $R^2$ | Pr(>$F$) | $R^2$ | Pr(>$F$) |
| Bulk soil | 0.2135 | 0.2934 | 0.2280 | 0.4283 |
| Rhizosphere | 0.1800 | 0.3378 | 0.3645 | 0.0005 |
| Artificial rhizosphere | 0.3690 | 0.0005 | 0.2862 | 0.0395 |
| Endosphere | 0.2402 | 0.3838 | 0.3208 | 0.1040 |

differential relative abundances as compared with control treatments (Welch's *t*-test, $P <$ 0.05, Fig. 2 and 3). These findings were further validated by direct bioassay, as explained below.

For bacteria, differential relative abundances were detected in the artificial rhizosphere compartment after amendments of 100 µM 1,8-cineole (13 OTUs) (Fig. 2A), 100 µM (3 OTUs) and 200 µM linalool (32 OTUs) (Fig. 2B and C), and 100 µM nerolidol (7 OTUs) (Fig. 2D) compared with water control. *Novosphingobium* was significantly and positively impacted by 1,8-cineole, but not by other terpenes. On the other hand, *Pseudomonas* and *Dechloromonas* were both significantly and positively impacted by 100 µM linalool and 100 µM nerolidol treatments. At higher concentrations of linalool (200 µM), *Azovibrio, Panacagrimonas, Thauera,* and *Variovorax* had relative abundances ~5% greater than controls.

For fungi, significant differential relative abundances were detected in the artificial rhizosphere compartment after amendments of 100 µM 1,8-cineole (13 OTUs) (Fig. 3A), 100 µM (4 OTUs) and 200 µM linalool (10 OTUs) (Fig. 3B and C), and 100 µM nerolidol (2 OTUs) (Fig. 3D) compared with water control. *Penicillium, Fusarium,* and *Orbilia* were significantly and the most positively impacted by 1,8-cineole, while *Mortierella, Tetracladium,* and *Gibellulopsis* OTUs were negatively impacted by this monoterpene. *Orbilia* was also significantly and positively impacted by 100 µM linalool. However, at higher levels of linalool (200 µM), *Fusarium* OTUs were the most positively impacted fungal OTUs in the community.

Indicator bacterial and fungal OTUs were also identified in the artificial rhizosphere for each terpene treatment (Tables S3 and S4). A total of 95 bacterial and 19 fungal OTUs were found to be indicative of the terpene treatments. Some OTUs were detected in both analyses including fungal OTUs identified as *Cadophora* sp., *Penicillium simplicissimum*, and *Tetracladium* sp., and bacterial OTUs identified as *Ahniella* sp., *Dechloromonas* spp., *Pedosphaeraceae* Ellin517 sp., *Panacagrimonas* sp., *Polycyclovorans* sp., *Azovibrio* sp., *Sphingomonas* spp., and *Thauera* sp. To validate the terpene effects on microbial growth, these community-level results were used to guide the selection of the bacterial and fungal isolates for direct screening for growth responses on terpene-amended media as described below.

## Microbial isolates from sorghum roots

In this study, a total of 68 fungal isolates were generated from sorghum roots, of which 13 fungal OTUs were found to have 100% sequences match to those detected as responsive taxa in the community data set (Table 3). For bacteria, 72 pure isolates were obtained from the sorghum roots. Among the 72 bacterial isolates, 30 were found to have 100% sequences match with the OTUs (Table 3). We used these isolates that had 100% sequence that match to responsive taxa, as well as other isolates representing genera as either the differential relative abundance or the indicator species from the microbiome analyses, in the plate assay to validate the effects of these terpenes on the growth of specific bacterial and fungal species by subculturing selected isolates on growth media amended with the same terpene treatments as described above.

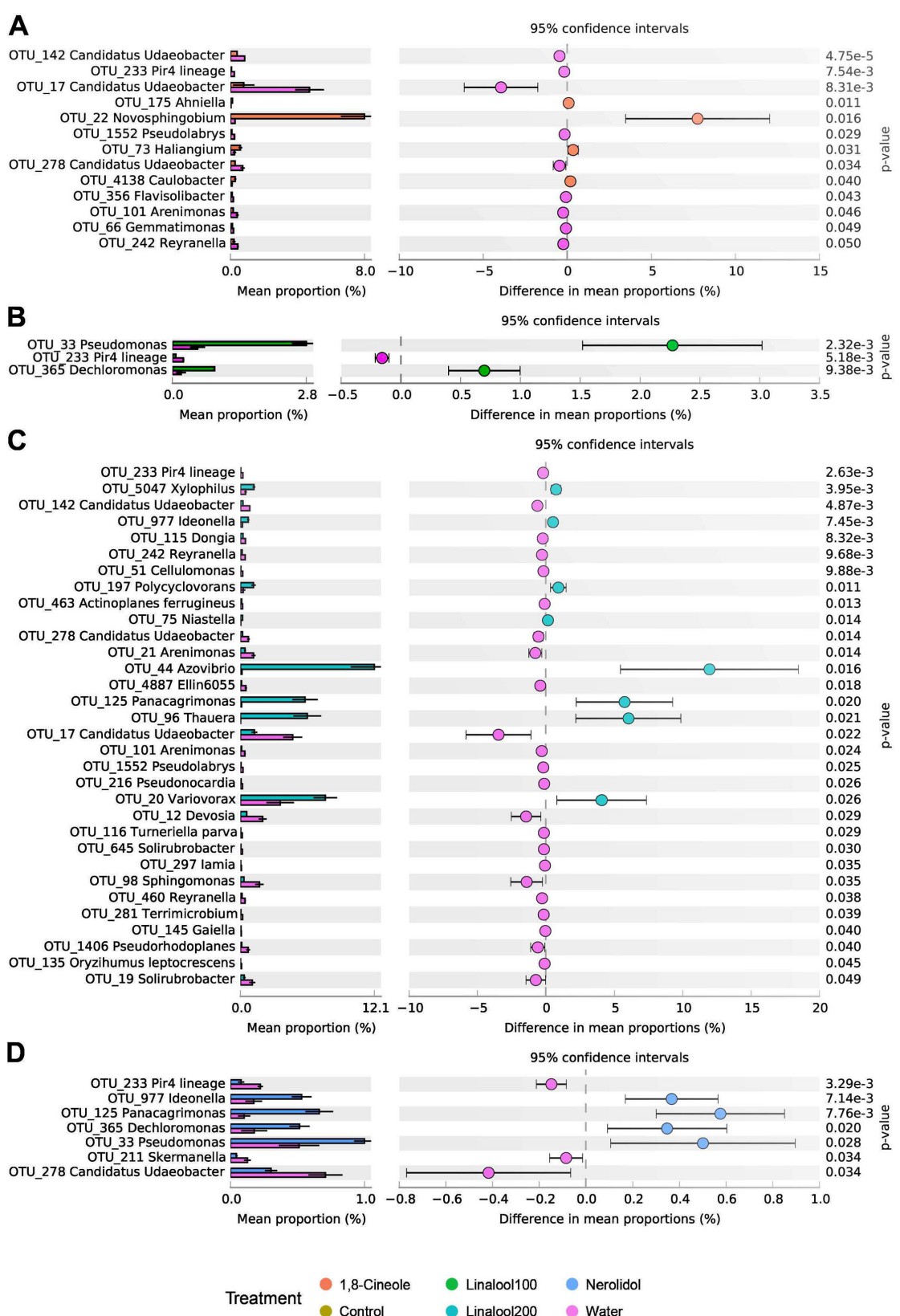

**FIG 2** Differential relative abundance of bacterial OTUs between terpene-treated and -untreated artificial rhizosphere in sorghum including 1,8 Cineole (A), Linalool 100 (B), Linalool 200 (C) and Nerolidol (D). Only the OTUs with 0.1% relative abundance were selected and statistically tested with Welch's *t*-test.

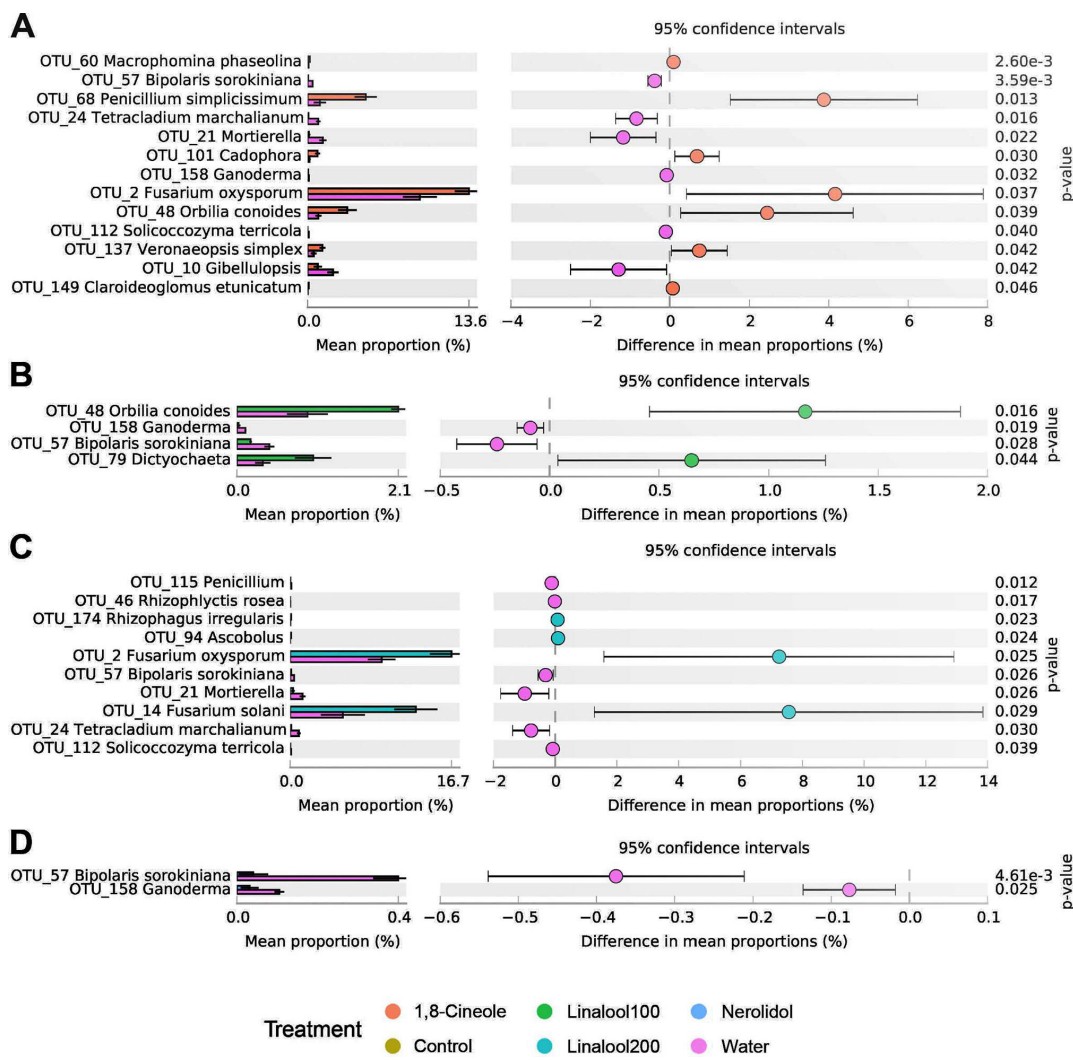

**FIG 3** Differential relative abundance of fungal OTUs between terpene-treated and -untreated artificial rhizosphere in sorghum including 1,8 Cineole (A), Linalool 100 (B), Linalool 200 (C) and Nerolidol (D). Only the OTUs with 0.1% relative abundance were selected and statistically tested with Welch's *t*-test.

## Terpene screening of select bacterial and fungal pure culture isolates/OTUs

For bacteria, an initial growth rate pre-screen of 63 bacterial isolates was carried out in the presence of 200 µM of 1,8-cineole, linalool, nerolidol, or ½ dilution of Tryptic Soy (50TSB) broth medium as control (Fig. S5) for 25 h. From these, based on differential growth responses toward the terpenes, a subset of 12 isolates was selected for further replicated growth assays on media amended with 100 µM or 200 µM of 1,8-cineole, linalool, and nerolidol for an extended duration up to 48 h (Fig. 4). We observed a slight growth promotion of *Bacillus_sp._5D* and delayed growth response in growth rate of isolate *Bacillus_sp_4A* under the presence of nerolidol at 100 µM. Isolates *Bacillus_sp_4A, Ferrovibrium_sp_7F, Lysinibacillus_sp_4B*, and *Lysinibacillus_sp_9B* show inhibited growth in the presence of nerolidol at 200 µM final concentration.

Of the nine fungal isolates tested, all were inhibited by nerolidol to a certain extent, with some (*Clonostachys* and *Humicola*) being more inhibited by the smaller 100 µM dosage than the higher 200 µM of nerolidol. In fact, the 200 µM of nerolidol appeared to promote the growth of *Clonostachys* (Fig. 5). Yet, other taxa (*Penicillium* and *Periconia*) were only significantly inhibited by the 200 µM dose of nerolidol rather than the lower concentration of 100 µM. 1,8-cineole had an inhibitory effect on *Orbilia* at both tested concentrations but had a promotive effect at 100 µM on *Penicillium* and *Periconia*. These

**TABLE 3** Fungal and bacterial isolates used in the terpene-amended plate assay that had matching sequences with amplicon-sequencing OTUs

| Fungal isolates | Genbank accession | Matching fungal OTUs | % similarity | Sequence length | Taxonomy |
|---|---|---|---|---|---|
| GLBRC1136 | OQ434097 | OTU_2 | 100 | 229 | *Fusarium* sp. |
| GLBRC1195 | OQ434156 | OTU_3 | 100 | 229 | *Exophiala* sp. |
| GLBRC1185 | OQ434146 | OTU_4 | 100 | 229 | *Humicola* sp. |
| GLBRC1135 | OQ434096 | OTU_8 | 100 | 229 | *Exophiala* sp. |
| GLBRC1140 | OQ434101 | OTU_17 | 100 | 229 | *Periconia* sp. |
| GLBRC1197 | OQ434158 | OTU_19 | 100 | 229 | *Penicillium* sp. |
| GLBRC1134 | OQ434095 | OTU_25 | 100 | 195 | *Serendipita* sp. |
| GLBRC1161 | OQ434122 | OTU_28 | 100 | 229 | *Clonostachys* sp. |
| GLBRC1142 | OQ434103 | OTU_35 | 100 | 229 | *Poaceascoma* sp. |
| GLBRC1156 | OQ434117 | OTU_75 | 100 | 229 | *Trichoderma* sp. |
| GLBRC1167 | OQ434128 | OTU_111 | 100 | 229 | *Rhodotorula* sp. |
| GLBRC1153 | OQ434114 | OTU_119 | 100 | 229 | *Cordycepts* sp. |
| GLBRC1143 | OQ434104 | OTU_877 | 99 | 226 | *Mortierella alpina* |
| GLBRC1178 | OQ434139 | OTU_111 | 100 | 229 | *Rhodotorula* sp. |
| GLBRC1199 | OQ456128 | OTU_48 | 99.563 | 229 | *Orbilia conoides* |
| **Bacterial isolates** | **Genbank accession** | **Bacterial OTUs** | **% similarity** | **Sequence length** | **Taxonomy** |
| MML_MYC-1B | OQ509753 | OTU_177 | 100 | 253 | *Ralstonia pickettii* |
| MML_MYC-1D | OQ509754 | OTU_177 | 100 | 253 | *Ralstonia pickettii* |
| MML_MYC-1F | OQ509755 | OTU_177 | 100 | 253 | *Ralstonia pickettii* |
| MML_MYC-1G | OQ509786 | OTU_20 | 100 | 253 | *Variovorax* sp. |
| MML_MYC-2D | OQ509719 | OTU_74 | 100 | 253 | *Bacillus* sp. |
| MML_MYC-2F | OQ509720 | OTU_74 | 100 | 253 | *Bacillus* sp. |
| MML_MYC-3C | OQ509756 | OTU_177 | 100 | 253 | *Ralstonia pickettii* |
| MML_MYC-3E | OQ509757 | OTU_177 | 100 | 253 | *Ralstonia pickettii* |
| MML_MYC-3H | OQ509778 | OTU_59 | 100 | 253 | *Streptomyces* sp. |
| MML_MYC-4C | OQ509777 | OTU_59 | 100 | 253 | *Streptomyces* sp. |
| MML_MYC-4F | OQ509758 | OTU_177 | 100 | 253 | *Ralstonia pickettii* |
| MML_MYC-4G | OQ509759 | OTU_177 | 100 | 253 | *Ralstonia pickettii* |
| MML_MYC-5A | OQ509730 | OTU_50 | 100 | 240 | *Caulobacter* sp. |
| MML_MYC-5B | OQ509760 | OTU_177 | 100 | 253 | *Ralstonia pickettii* |
| MML_MYC-5D | OQ509717 | OTU_62 | 100 | 253 | *Bacillus* sp. |
| MML_MYC-5F | OQ509718 | OTU_62 | 100 | 253 | *Bacillus* sp. |
| MML_MYC-5G | OQ509721 | OTU_74 | 100 | 253 | *Bacillus* sp. |
| MML_MYC-6D | OQ509761 | OTU_177 | 100 | 159 | *Ralstonia pickettii* |
| MML_MYC-6G | OQ509787 | OTU_20 | 100 | 253 | *Variovorax* sp. |
| MML_MYC-7C | OQ509767 | OTU_1546 | 100 | 253 | *Sphingoaurantiacus* sp. |
| MML_MYC-7G | OQ509734 | OTU_609 | 100 | 253 | *Ferrovibrio* sp. |
| MML_MYC-7H | OQ509768 | OTU_1546 | 100 | 253 | *Sphingoaurantiacus* sp. |
| MML_MYC-8A | OQ509769 | OTU_1546 | 100 | 253 | *Sphingoaurantiacus* sp. |
| MML_MYC-8F | OQ509780 | OTU_428 | 100 | 253 | *Tardiphaga* sp. |
| MML_MYC-8H1 | OQ509740 | OTU_363 | 100 | 190 | *Methylobacterium* sp. |
| MML_MYC-9E | OQ509765 | OTU_6211 | 100 | 253 | *Roseomonas* sp. |
| MML_MYC-10A | OQ509741 | OTU_363 | 100 | 253 | *Methylobacterium* sp. |
| MML_MYC-10C | OQ509746 | OTU_515 | 100 | 253 | *Nonomuraea* sp. |
| MML_MYC-10F | OQ509739 | OTU_9562 | 100 | 211 | *Mesorhizobium* sp. |
| MML_MYC-11A | OQ509729 | OTU_50 | 100 | 253 | *Caulobacter* sp. |

growth promotion relationships did decrease in significance for *Penicillium* and *Periconia* as time passed and the isolates filled the plates (Fig. S4). Linalool at 100 µM had a mild but significant growth promotion effect in *Mortierella*, but an inhibitory effect for *Orbilia*.

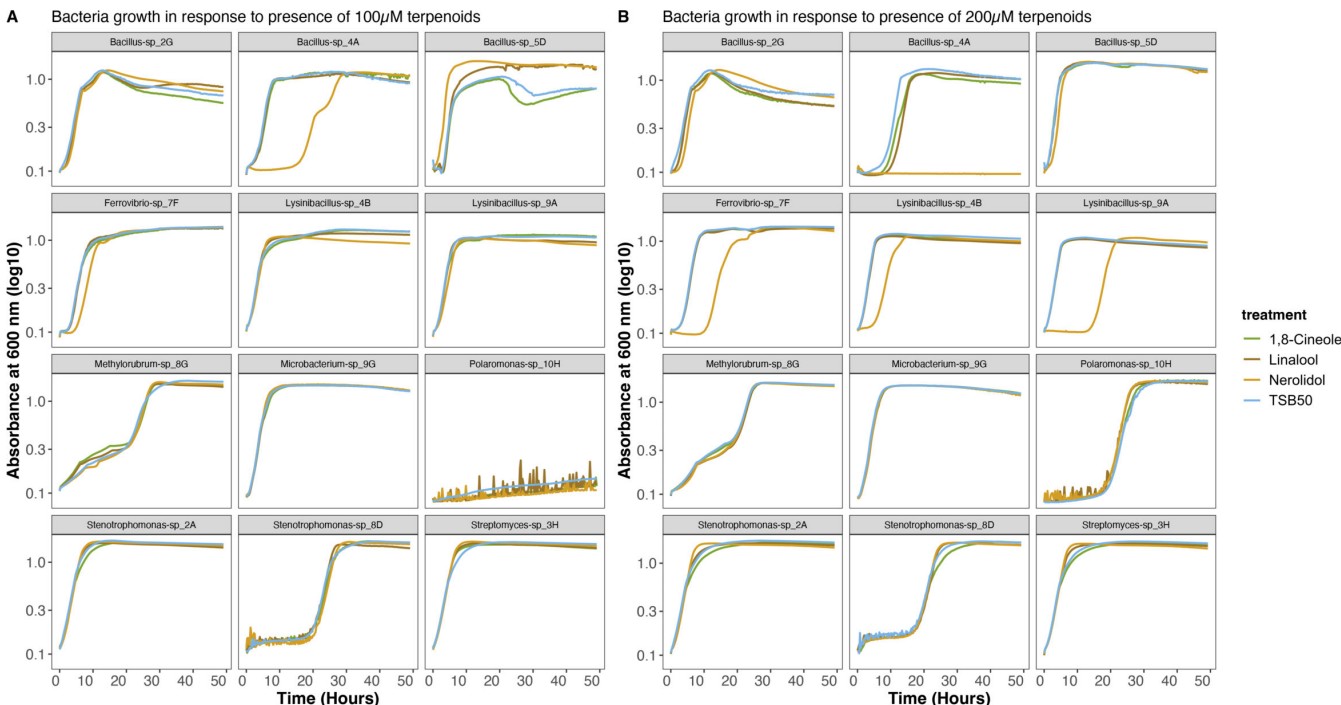

**FIG 4** Replicate growth curves of selective sorghum root bacterial isolates grown in 0.5X TSB compared with those grown in 0.5X TSB amended with 1,8-cineole, linalool, or nerolidol at 100 µM (A) and 200 µM for extended time up to 50 h measured by cell density at $OD_{600}$.

## GC-MS analysis of terpenoids in the soil

Soil samples were harvested from the artificial rhizosphere 24 h post-treatment and analyzed by GC-MS at 35–250 m/z. It was not possible to detect linalool, 1,8-cineole, or nerolidol. This supports that no significant build-up of terpenes in soil occurred over the course of the experiment and that the treatments were transient in nature due to the volatility of the terpenoids. Soil terpenoids are known to be subject to complete loss through vertical diffusion. Nerolidol and linalool were earlier shown to readily diffuse in soil and sand and could not be detected beyond 1.5 and 6 cm from the injection site, respectively (44).

## DISCUSSION

Terpenoids play important roles in plant adaptation and their responses to the environment including abiotic and biotic interactions. In this study, we investigated the composition and abundance of the belowground sorghum root and soil-associated microbiome in response to the addition of commercially available terpenoids that have previously been detected in root metabolomes. Three terpenoids were supplemented experimentally through an artificial root matrix in the sorghum rhizosphere to mimic the release from the root system, and changes to bacterial and fungal microbiomes in control and experimental treatments were assessed. Significant and differential impacts of sesquiterpenes and monoterpenes on fungal and bacterial communities and pure culture bioassays were detected.

## Localized impact of terpenes on microbial communities

Results from this experiment of adding terpenes into soils with growing sorghum plants demonstrated that the application of nerolidol, linalool, and 1,8-cineole through the artificial rhizosphere had a localized effect on the bacterial and fungal microbiome associated with this compartment and was also found to modulate the rhizosphere fungal community. Interestingly, the localized terpenoid enrichment had differential impacts between microbial guilds, specific terpenes, and terpene concentrations.

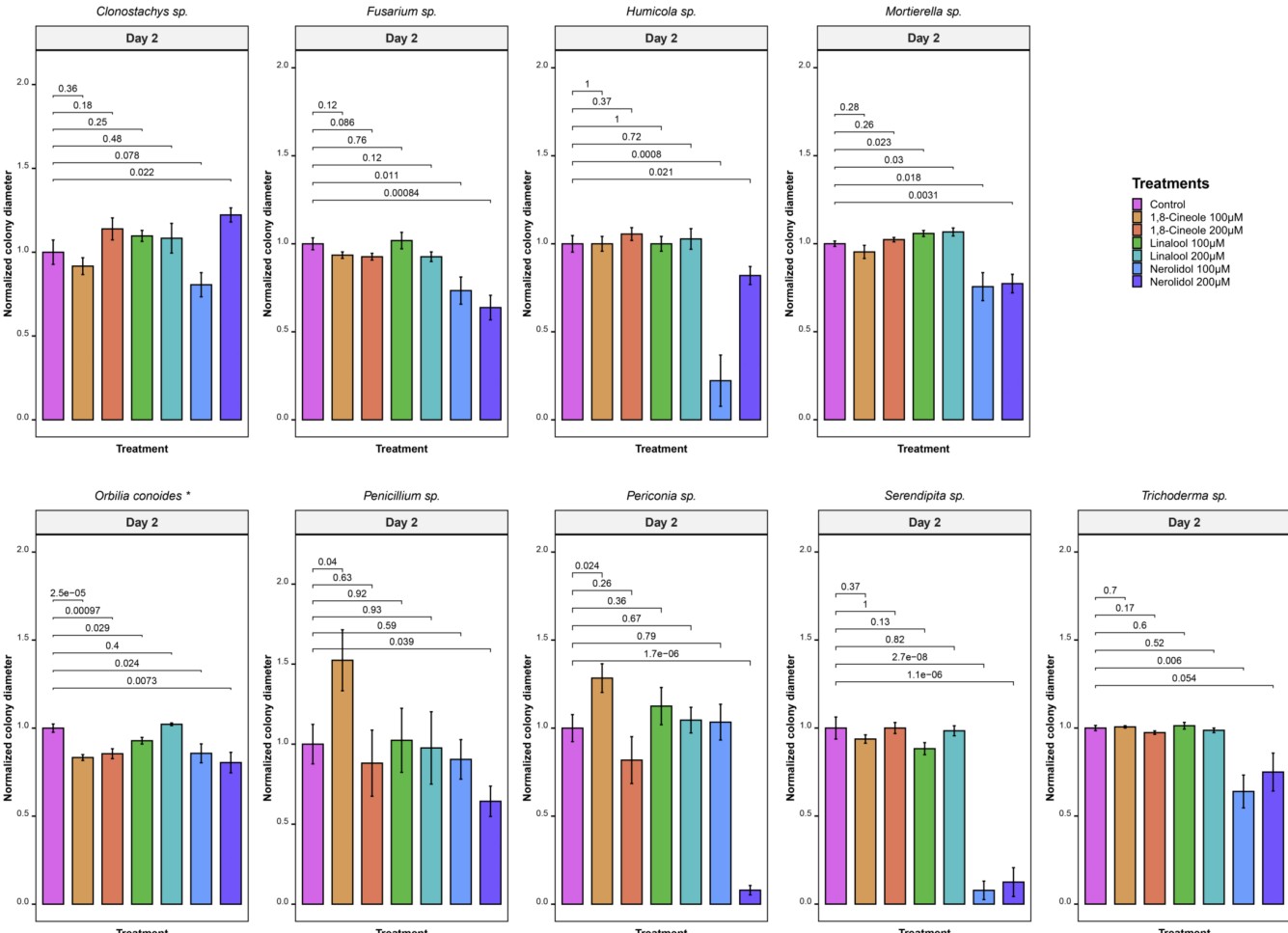

**FIG 5** Sorghum root fungal isolate growth assay performed on 0.1X PDA containing 1,8-cineole, linalool, and nerolidol at 100 and 200 µM, and compared with 0.1X PDA control. The bars represent the colony radius normalized against the control after 48 h of incubation, and the numbers above represent the *P*-value derived from Student's *t*-test comparing each treatment with the control.

In the case of bacteria, the relative abundance of one *Pseudomonas* OTU in our study was found to be enriched by both linalool and nerolidol at the concentrations of 100 µM. This is consistent with *Pseudomonas* spp. that can use linear terpenoids such as linalool as a sole carbon source for growth (45). The catabolic acyclic terpene utilization pathway in *Pseudomonas* is typically present in a gene cluster and proceeds via oxidoreductase reactions and CoA-activation into the central leucine/isovalerate utilization pathways. Intriguingly, linalool has demonstrated anti-microbial activity against the pathogenic *P. fluorescens* and *P. aeruginosa*. Still, given that some *Pseudomonas* species (and strains) are known for their various plant-growth-regulating functions such as nutrient cycling and pathogen suppression, others can be pathogenic to plants, so the impact of this on the plant is difficult to ascertain (46).

Another interesting result was the enriched relative abundance of a *Novosphingobium* OTU, only detected in the monoterpene 1,8-cineole treatments. Although the specific ecology of this OTU is unclear, *Novosphingobium* is generally considered to be plant-growth-promoting rhizobacteria (47). Similarly, an *Azovibrio* OTU was the most enriched OTU (>12%) in the lineole treatment but only at the highest (200 µM) concentration tested. *Azovibrio* are plant-associated bacteria and are generally considered to be beneficial given their ability to express nitrogenase (NifH) genes and to fix atmospheric nitrogen in grasses; they can also be associated with arbuscular mycorrhizal fungal spores (48–50). By contrast, the relative abundance of the *Pirellulales*-related Pir4 clade

was consistently reduced by the addition of terpenoids. While the ecological function of these bacteria is unclear, they have been reported as keystone taxa in rehabilitated soils and also as disease organisms of macroalgae (51, 52). Unfortunately, none of our isolates matched *Novosphingobium, Azovibrio,* or *Pirellulales* OTUs so we were unable to follow up directly on their responses through bioassays.

Similarly to what was observed for the bacterial community, the localized terpenoid enrichment had differential impacts between fungal guilds and varied between the specific terpenes and terpene concentrations tested. *Fusarium* spp. were among the most responsive fungal taxa and showed an increase in abundance of >4% in the 1,8-cineole and linalool (200 µM) monoterpene treatments. *Fusarium* is a large genus with over 1,000 species that are considered to be ubiquitous in the environment. Although there are several well-known soil-borne pathogenic species capable of causing diseases across a wide variety of crops (53), plant-beneficial strains exist, even in phytopathogenic species complexes(54–56).

Another fungal OTU that responded positively to 1,8-cineole belonged to *Penicillium*. While many species of *Penicillium* are known to be post-harvest food pathogens, most *Penicillium* species are considered to be beneficial to growing plants given their ability to solubilize phosphate and to biosynthesize siderophores, plant-growth regulators, and diverse secondary metabolites (57). *Orbilia conoides* was enriched in the monoterpene 1,8-cineole and linalool (100 µM) treatments. This species is also considered to be beneficial to plants as a biocontrol agent, given its ability to trap and kill nematodes and to solubilize phosphate in soils (58–60). In the case of arbuscular mycorrhizal fungi that are obligate plant symbionts, linalool (200 µM) had a positive impact on the abundance of *Rhizophagus,* and 1,8-cineole had a positive impact on the abundance of *Claroideoglomus*. The enrichment of arbuscular mycorrhizal fungi *Rhizophagus* in linalool (200 µM) treatment could have facilitated the nutrient availability for sorghum and contributed to the observed healthy foliar growth.

In contrast to the increases in fungal relative abundance discussed above, decreases were also observed. For instance, 1,8-cineole caused a decrease in the abundance of the plant-growth-promoting *Mortierella* (61). It is also noteworthy that the relative abundance of *Ganoderma* was consistently reduced by all terpenes tested. *Ganoderma* is known for producing bioactive triterpenes and lignin-degrading enzymes that facilitate the breaking down of plant litter and carbon cycling in the soil and wood (62–64). Similarly, *Bipolaris sorokiniana* has the capacity to produce a rich spectrum of phytotoxic sesquiterpenoids and other terpene-derived products (65), and its growth was reduced by all three of the terpenoids and at both linalool concentrations. Some isolates of *B. sorokiniana* are known to cause diseases on cereal crops, including *S. bicolor*; thus, their reduction could be considered positive (66, 67). Accordingly, the increase in belowground biomass in the nerolidol treatment could be due suppression of plant antagonistic microbes. However, we acknowledge that the variation of root sampling for microbiome analysis and isolation, although minor in relative to the whole root system, could also have caused the differences in belowground biomass.

Overall, more OTUs were found to be affected by linalool (200 µM) and 1,8-cineole than by linalool (100 µM) or nerolidol. This was true for bacteria and fungi and aligned well with what was observed in impacts on bacterial and fungal richness (Fig. S6). While we may have expected that a higher concentration of linalool (200 µM) would have a greater impact on the microbial community compared to a lower concentration (100 µM), we found it particularly interesting that it was not the same taxa that were responding to the different levels indicating dose-responses differ between organisms and calling into question the distribution of such dose-response relationships (68). However, we did not expect that the sesquiterpene nerolidol would have fewer effects on the bacterial and fungal relative abundance in a community setting compared to the monoterpene treatments, as nerolidol is well known for its antibiotic activities (69, 70). One potential explanation for this observation is that the amplicon sequencing only generates relative abundance data which does not reflect the overall microbial

abundance. Nerolidol could have a general inhibition effect and reduce the overall microbial load but was not captured in the differential relative abundance analyses, which merely reflect the differences in sensitivity of microbes toward the nerolidol inhibition.

While terpene applications were evident in the artificial rhizosphere samples, and to a lesser extent in the fungal rhizosphere of sorghum, overall, the terpene application did not mediate the microbiome structure in bulk soil or root endosphere. There are many factors that could account for this. First, we did not detect any significant build-up of terpenes in the soil over the course of the experiment indicating the treatment effects were highly localized, as the terpenes were applied belowground through the artificial roots. Second, it is known that root endophytes are resilient to abiotic disturbances given that the root endosphere is a highly selective niche composed of microbes that are well-adapted to and shaped by long-term host-microbe co-evolution (71, 72). The fact that terpene treatments impacted fungal but not bacterial rhizosphere communities shows that the rhizosphere fungal community is more sensitive to terpene disturbance. This effect of terpenes on the fungal community likely comes from three main routes: (i) direct terpene mediation on the fungal growth, (ii) indirect mediation through regulating plant metabolism, or (iii) a combination of the two. It is also possible that the terpenes affected the metabolism of bacteria without changing their relative abundance which then mediated the observed fungal community shifts.

## Impact of terpenes on the growth of individual fungi and bacteria

Bioassays using bacteria and fungi isolated from the experiment described above allowed us to validate results from the mesocosms more directly using microbial pure cultures. Interestingly, among the three terpenes tested, bacterial isolates were only responsive to nerolidol, and all the responsive bacteria were inhibited by it with the sole exception being *Bacillus* sp. 5D, which was promoted by nerolidol (Fig. S5). Those showing growth inhibition by nerolidol in our study include those under genera *Bacillus*, *Ferrovibrio,* and *Lysinibacillus*. For *Bacillus*, this aligns with previous findings showing evidence of the inhibitory effect of different nerolidol forms on its growth (69, 70). However, there were no documents on the inhibitory effect of nerolidol on *Ferrovibrio* and *Lysinibacillus*. Furthermore, except for *Bacillus* 4A in the nerolidol 200 µM assay, the inhibition was temporary with all isolates eventually reaching the same population density as control treatments indicating nerolidol was temporarily biostatic but not biocidal. We believe this is the first study showing a growth promotional effect of nerolidol on *Bacillus*.

Fungal bioassay results were more variable than those of bacteria, but similarly, nerolidol tended to have an inhibitory response. This was true with the exception of *Clonostachys* which was inhibited at the 100 µM but stimulated at the 200 µM concentration. This raises the interesting question of whether *Clonostachys* is capable of catabolizing nerolidol and using it as a carbon source. Although such activity on terpenoids has not yet been identified, it was earlier demonstrated that *Clonostachys* can oxidatively biotransform plant-specialized metabolites of the alkaloid class through a cytochrome P450-based mechanism (73).

For *Humicola*, *Serendipita,* and *Trichoderma,* 100 µM nerolidol treatments were more inhibitory to growth than were treatments with 200 µM. By contrast, *Fusarium*, *Penicillium,* and *Periconia* showed greater inhibition at the higher 200 µM concentration. Nerolidol is known for its antibiotic activity (23), and even some fungi have been shown to biosynthesize nerolidol to suppress the growth of other fungi in their surroundings (22). For instance, *Trichoderma* spp., *F. culmorum*, *F. xyrophilum,* and *Schizophllum commune* biosynthesize different forms of nerolidol (22, 69, 74). Yet, *Trichoderma* and *Fusarium* isolates tested in our study were suppressed by nerolidol at both concentrations. Others have also reported that *Trichoderma* species can be suppressed by the application of terpenes despite the fact that these same organisms can biosynthesize terpenes, including nerolidol (75). While the growth inhibition of nerolidol was

temporary and most of the fungi tested recovered over time as the terpenes likely volatilized, *Humicola* and the plant-growth-promoting *Serendipta* isolates did not recover and stayed stunted in growth (76). This is particularly interesting given the fact that in *Serendipita* colonized tomato plants, terpene synthase genes involved in nerolidol biosynthesis are upregulated in the leaves, but not the roots (77).

Terpene 1,8-cineole had a stimulatory effect on *Penicillium* and *Periconia*, but only at the 100 µM concentration. This response is consistent with the response that was observed for *Penicillium* in the community analysis. However, it was unexpected that out of the nine fungal isolates tested *Orbilia conoides* was the only one whose growth was inhibited by 1,8-cineole given that this same species had a greater relative abundance in the community analysis. Likewise, the monoterpene linalool had a slight but significant growth stimulatory impact on *Mortierella alpina* at both concentrations tested, yet its relative abundance was significantly decreased in the community analysis in the 200 µM linalool treatment. Although *F. oxysporum* and *F. solani* were recently shown to be highly sensitive to linalool, the *Fusarium* isolate that we tested was not sensitive (78). These results highlight the challenges in extrapolating results between scales and may indicate that indirect effects contribute to changes in relative abundance in the community setting. Again, the compositional nature of the method used to study the microbial community re-emphasizes the importance of further validation.

Together, these findings of fungal and bacterial strains that are promoted by nerolidol may have ecological implications as they not only can thrive among nerolidol-sensitive microbes, but they may be crucial in maintaining the microbial taxonomic or functional diversity in a nerolidol-enriched environment. For application, these microbes can be used as a growth promoter/facilitator for other microbes or even as bioremediators in a nerolidol-supplemented environment.

We must acknowledge that the effect of terpenes on the microbial community was not always consistent with the response of the terpenes on single isolates. There may be several reasons for this. First, the growth environments were vastly different between soil and refined growth media. The substrate, chemical, and biological complexities of soils are much greater than that of the growth media, and labile carbon availability is also much lower in the soil environment. The substrate properties greatly affect the biochemical process in the microbes and their capacity in xenobiotic degradation and metabolism (79–81). Second, there is a lack of microbe-microbe and plant-microbe interactions in plate bioassays. For example, *Fusarium* can deactivate the antimicrobial activities of nerolidol by hydrating it into caparrapidiol, and this could benefit, or harm, microbes nearby (82). In addition, plants can catalyze oxidative cleavage yielding norterpenoids, including hormones, signaling molecules, and the bioactive dimethyl-nonatriene, which can disrupt the microbiome of an insect pest (83–85). These are just some examples of how plant-microbe-terpene interactions change the microbial composition dynamics within the artificial root community, which can mask the direct terpene effects observable when tested on a single microbial isolate.

## Applications of terpenes in agriculture and biotechnology

While most current studies have focused on modifying the plant terpenoid profiles, and subsequently investigating the effect on the plant microbiome, the transformation of plant-associated microbes provides the possibility of modifying microbes rather than plants to produce various terpenoids. The biosynthetic pathways to the terpenes used in this study are fully elucidated. Many bacteria and fungi have shown great promise for genetic transformation opening the possibility for biotechnological applications with terpenes. From our research, some relevant targets and approaches may include *Bacillus* (86)*, Pseudomonas* (87), *Serendipita* (11, 88), *Fusarium* (89, 90), *Mortierella* (91, 92), and *Trichoderma* (93). Importantly, some taxa including those of *Fusarium* and *Serendipita* are able to produce sesquiterpenoids and therefore already have a genetic background to possibly be modified to access relevant plant-beneficial terpenoids. In summary, this study highlights the complexity of plant-microbe-terpene interactions, and how

community responses may mask the direct effects observable in single microbial bioassays. Overall, this work opens avenues and targets for the engineering of terpene metabolisms in key plant microbiome guilds to improve sustainable agriculture and bioenergy production.

## ACKNOWLEDGMENTS

We are grateful to Rosanne Healy and Matthew E. Smith for useful discussion and resources on *Orbilia*, and Lisa Tiemann and Violeta Matus Acuna for providing nematodes for baiting of the nematode-trapping fungus *Orbilia*. We thank Reid Longley and Stacey Vanderwulp for assistance in obtaining field soils used in this study. We thank Sasha Kravchenko, Maxwell Oerther, Tayler Ulbrich, and Maik Lucas for lending and providing assistance in assembling and using the rhizoboxes for this study. Michigan State University occupies the ancestral, traditional, and contemporary Lands of the Anishinaabeg—Three Fires Confederacy of Ojibwe, Odawa, and Potawatomi peoples. The University resides on Land ceded in the 1819 Treaty of Saginaw.

This material is based upon work supported by the Great Lakes Bioenergy Research Center, U.S. Department of Energy, Office of Science, Office of Biological and Environmental Research under Award Numbers DE-SC0018409 and DE-FC02-07ER64494.

M.-Y.C.: conceived and performed research and data analysis, created figures and tables, and wrote the manuscript. T.B.A.: conceived and performed research and data analysis, created figures and tables, and wrote the manuscript. M.E.M.L.: performed research and data analysis, created figures and tables, and edited the manuscript. N.B.: performed research and data analysis, created figures and tables, and edited the manuscript. A.S.: performed research and data analysis, created figures and tables, and edited the manuscript. N.M.: performed research and data analysis, created figures and tables, and edited the manuscript. A.S.: conceived of experimental approaches and edited the manuscript. B.H.: conceived of experimental approaches, wrote and edited the manuscript. G.M.B.: conceived of experimental approaches and wrote and edited the manuscript.

## AUTHOR AFFILIATIONS

[1]Department of Plant Soil and Microbial Sciences, Michigan State University, East Lansing, Michigan, USA
[2]Great Lakes Bioenergy Research Center, Michigan State University, East Lansing, Michigan, USA
[3]Department of Plant Biology, Rutgers University, New Brunswick, New Jersey, USA
[4]Department of Biochemistry and Molecular Biology, Michigan State University, East Lansing, Michigan, USA
[5]Department of Microbiology and Molecular Genetics, Michigan State University, East Lansing, Michigan, USA
[6]Research Group on Bacterial Efflux and Environmental Resistance, CNRS, INRAe, École Nationale Véterinaire de Lyon and Université Lyon 1, Université de Lyon, Villeurbanne, France

## AUTHOR ORCIDs

Ming-Yi Chou http://orcid.org/0000-0001-8210-2577
Ashley Shade http://orcid.org/0000-0002-7189-3067
Bjoern Hamberger http://orcid.org/0000-0003-1249-1807
Gregory Bonito http://orcid.org/0000-0002-7262-8978

## FUNDING

| Funder | Grant(s) | Author(s) |
| --- | --- | --- |
| U.S. Department of Energy (DOE) | DE-SC0018409 | Ashley Shade |

| Funder | Grant(s) | Author(s) |
|---|---|---|
| | | Bjoern Hamberger |
| | | Gregory Bonito |
| U.S. Department of Energy (DOE) | DE-FC02-07ER64494 | Ashley Shade |
| | | Bjoern Hamberger |
| | | Ming-Yi Chou |
| | | Marco E. Mechan Llontop |
| | | Trine B. Andersen |
| | | Nick Beculheimer |
| | | Alassane Sow |
| | | Nick Moreno |
| National Science Foundation (NSF) | 1737898 | Bjoern Hamberger |
| | | Gregory Bonito |
| | | Alassane Sow |

## AUTHOR CONTRIBUTIONS

Ming-Yi Chou, Conceptualization, Data curation, Formal analysis, Methodology, Validation, Writing – original draft, Writing – review and editing | Trine B. Andersen, Conceptualization, Data curation, Formal analysis, Methodology, Validation, Writing – original draft, Writing – review and editing | Marco E. Mechan Llontop, Formal analysis, Methodology, Validation, Visualization, Writing – original draft, Writing – review and editing | Nick Beculheimer, Data curation, Formal analysis, Visualization, Writing – review and editing | Alassane Sow, Formal analysis, Methodology, Validation, Writing – review and editing | Nick Moreno, Formal analysis, Methodology, Validation, Visualization, Writing – review and editing | Ashley Shade, Conceptualization, Funding acquisition, Methodology, Project administration, Supervision, Writing – review and editing | Bjoern Hamberger, Conceptualization, Funding acquisition, Investigation, Project administration, Supervision, Visualization, Writing – original draft, Writing – review and editing | Gregory Bonito, Conceptualization, Investigation, Project administration, Supervision, Writing – original draft, Writing – review and editing

## DATA AVAILABILITY

The fungal ITS and partial LSU sequences have been submitted to NCBI under the accession numbers OQ434094-OQ434159, OQ456128, and OQ469489. Bacterial sequences have been submitted to NCBI under the accession numbers OQ509716–OQ509787. All demultiplexed amplicon sequencing reads have been submitted to the NCBI Sequence Read Archive under BioProject number PRJNA933671.

## ADDITIONAL FILES

The following material is available online.

### Supplemental Material

**Supplemental Figures (Spectrum01332-23-s0001.docx).** Supplemental Figures S1 to S6.
**Table S1 (Spectrum01332-23-s0002.docx).** Full PERMANOVA results of the terpene amendment effects.
**Table S2 (Spectrum01332-23-s0003.xlsx).** Phenotypes and growth parameters of sorghum grown with terpene amendment through artificial rhizosphere.
**Table S3 (Spectrum01332-23-s0004.xlsx).** Bacterial indicator OTUs for each terpene treatment in the artificial rhizosphere.

**Table S4 (Spectrum01332-23-s0005.xlsx).** Fungal indicator OTUs for each terpene treatment in the artificial rhizosphere.

Open Peer Review

**PEER REVIEW HISTORY (review-history.pdf).** An accounting of the reviewer comments and feedback.

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
