## [Reviewer comments · Microbiology Spectrum]

Microbiology Spectrum

Terpenes modulate bacterial and fungal growth and sorghum rhizobiome communities

Ming-Yi Chou, Trine Andersen, Marco Mechan-Llontop, Nick Beculheimer, Alassane Sow, Nick Moreno, Ashley Shade, Bjoern Hamberger, and Gregory Bonito

Corresponding Author(s): Gregory Bonito, Michigan State University

Review Timeline:

Submission Date:	March 28, 2023
Editorial Decision:	May 2, 2023
Revision Received:	May 23, 2023
Editorial Decision:	May 24, 2023
Revision Received:	May 26, 2023
Accepted:	July 5, 2023

Editor: Erik Hom

Reviewer(s): Disclosure of reviewer identity is with reference to reviewer comments included in decision letter(s). The following individuals involved in review of your submission have agreed to reveal their identity: Victoria Calatrava (Reviewer #1)

Transaction Report:

DOI: <https://doi.org/10.1128/spectrum.01332-23>

May 2, 2023

Dr. Gregory Bonito
Michigan State University
Plant Soil Microbial Sciences
East Lansing, MI 48823

Re: Spectrum01332-23 (Terpenes modulate bacterial and fungal growth and sorghum rhizobiome communities)

Dear Greg/Dr. Bonito:

Thank you for submitting your manuscript to Microbiology Spectrum. Your manuscript was reviewed by two experts and their comments are below; both suggest revisions.

With regard to Reviewer #2's comments, I look forward to your response. If indeed nerolidol solubility is a problem, you might consider excluding that data or labeling the solubility as saturated ($\leq 63 \mu\text{M}$) and mention that it was applied in excess. If you don't show the nerolidol data, then it would not make sense to include the associated sequencing data. Moreover, if the solubility of the other terpenes is a problem for the growth curves, please consider repeating those experiments.

When submitting the revised version of your paper, please provide (1) a brief summary of what you changed in your cover letter to me, (2) point-by-point responses to the issues raised by the reviewers as file type "Response to Reviewers," not in your cover letter, and (3) a PDF file that indicates the changes from the original submission (by highlighting or underlining the changes) as file type "Marked Up Manuscript - For Review Only". Please use this link to submit your revised manuscript - we strongly recommend that you submit your paper within the next 60 days or reach out to me. Detailed instructions on submitting your revised paper are below.

Link Not Available

Sincerely,

Erik Hom

Journals Department
Reviewer comments:

Reviewer #1 (Comments for the Author):

Terpenes are well known plant-derived products that affect plant development and adaptation, as well as biotic interactions with

insects and microbes. However, there is limited knowledge about their impact on the plant rhizosphere. This study focuses on the impact of three different terpenes on the rhizobiome by adding them exogenously using an artificial root system. The authors investigated the impact of terpenes on the relative abundance of bacteria and fungi. Additionally, they isolated a large number of bacterial and fungal species and study the impact of the different terpenes on their axenic growth to complement the microbiome data. The microbiome results showed that specific terpenes had varying impacts on specific bacterial and/or fungal taxa, and these were not always reproducible using single isolates. Dose-specific responses were also observed. These findings advance our knowledge of how terpenes impact the rhizosphere microbiome and could be useful for future applications to improve sustainable agriculture and bioenergy production. The results are well presented and discussed. I do not have any major issues, but do have a few minor suggestions and comments.

- Line 30: "Firmicutes" is a phylum and it should not be italicized. Please remove italics.
- Revise the text for unnecessary dash lines (e.g., line 55: "demon-strated"; line 58: "triter-pene", line 61: "re-por-ted"...)
- Line 317: By "rhizobium", do the authors mean rhizobiome instead?
- It seems that Tables S1 and S2 may have been mislabeled in lines 332 and 337, please revise.
- I recommend the use of either 1,8-cineole or cineole throughout the text for consistency.
- Line 533. Could the effect of terpenes on fungal community come also from an indirect effect by shifting the metabolism of the bacterial community, despite not affecting their relative abundance?
- Line 556. Are there any known enzymatic activities for nerolidol catabolism and are they found in *Clonostachys*?
- Line 562: Correct "Schizophllum" (*Schizophyllum*)
- Table 3: Correct "Taxaonomy" (Taxonomy)

Reviewer #2 (Public repository details (Required)):

The authors used NGS methods to identify fungi and bacteria and already supplied NCBI accession numbers.

Reviewer #2 (Comments for the Author):

Chou et al. investigated how terpenes influence the microbial community close to the roots of sorghum. While I find the paper well written and the topic interesting, there is one major design flaw for at least one of the terpenes. Nerolidol was used at concentrations of 100 (and later additionally at 200 μM), however the solubility in water (according to millipore/sigma) is only 0.014 g/L with a molecular mass of 222 g/mol the maximal solubility would be 63 μM , so the indicated concentrations are not possible. Moreover in the experiments where terpenes were predissolved (line 290) and added to autoclaved medium the concentration would have to be even higher to reach 100 or 200 μM . As no concentrations are given, I can not judge if the other 2 terpenes would also have been close or above their solubility in water. So at best only the results for nerolidol are questionable, at worst growth curves for the other 2 terpenes are also in question.

Minor comments would be to include information in the introduction about terpenes that are produced by sorghum, especially in the roots.

Line 116: Are the dimensions for one compartment or or all three together?

Line 126: How does the artificial root system look like, is there a reference or a vendor?

Line 194/195: What m/z range was used during detection?

Staff Comments:

Preparing Revision Guidelines

Please return the manuscript within 60 days; if you cannot complete the modification within this time period, please contact me. If you do not wish to modify the manuscript and prefer to submit it to another journal, please notify me of your decision immediately so that the manuscript may be formally withdrawn from consideration by Microbiology Spectrum.

Response to reviewers

The authors greatly appreciate the insightful comments provided by the editor and the two reviewers. Each point and concern raised by the reviewers is included below along with our responses. We feel the manuscript has been greatly improved due to the collective efforts from the editor and the reviewers.

Reviewer #1 (Comments for the Author):

Terpenes are well known plant-derived products that affect plant development and adaptation, as well as biotic interactions with insects and microbes. However, there is limited knowledge about their impact on the plant rhizosphere. This study focuses on the impact of three different terpenes on the rhizobiome by adding them exogenously using an artificial root system. The authors investigated the impact of terpenes on the relative abundance of bacteria and fungi. Additionally, they isolated a large number of bacterial and fungal species and study the impact of the different terpenes on their axenic growth to complement the microbiome data. The microbiome results showed that specific terpenes had varying impacts on specific bacterial and/or fungal taxa, and these were not always reproducible using single isolates. Dose-specific responses were also observed. These findings advance our knowledge of how terpenes impact the rhizosphere microbiome and could be useful for future applications to improve sustainable agriculture and bioenergy production. The results are well presented and discussed. I do not have any major issues, but do have a few minor suggestions and comments.

- Line 30: "Firmicutes" is a phylum and it should not be italicized. Please remove italics.

Response: Corrected.

- Revise the text for unnecessary dash lines (e.g., line 55: "demon-strated"; line 58: "triter-pene", line 61: "re-por-ted"...)

Response: Unnecessary marks were removed throughout.

- Line 317: By "rhizobium", do the authors mean rhizobiome instead?

Response: Yes, this is corrected and the text was edited.

- It seems that Tables S1 and S2 may have been mislabeled in lines 332 and 337, please revise.

Response: Texts were corrected.

- I recommend the use of either 1,8-cineole or cineole throughout the text for consistency.

Response: Acknowledged and confirmed corrections throughout.

- Line 533. Could the effect of terpenes on fungal community come also from an indirect effect by shifting the metabolism of the bacterial community, despite not affecting their relative abundance?

Response: The mediation of the fungal community by bacteria is possible. We added a sentence to complement our hypothesis in line 552-554.

• Line 556. Are there any known enzymatic activities for nerolidol catabolism and are they found in *Clonostachys*?

Response: While studies identified terpene production and gene clusters related to terpene synthases in *Clonostachys*, to our knowledge no enzymes or associated catabolic activity has been demonstrated in this genus. We clarified the capacity of *Clonostachys* to biotransform other plant specialized metabolites in line 572-576.

• Line 562: Correct "Schizopllum" (*Schizophyllum*)

Response: Corrected.

• Table 3: Correct "Taxaonomy" (Taxonomy)

Response: Corrected on Table 3.

Reviewer #2 (Comments for the Author):

Chou et al. investigated how terpenes influence the microbial community close to the roots of sorghum.

While I find the paper well written and the topic interesting, there is one major design flaw for at least one of the terpenes. Nerolidol was used at concentrations of 100 (and later additionally at 200 μM), however the solubility in water (according to millipore/sigma) is only 0.014 g/L with a molecular mass of 222 g/mol the maximal solubility would be 63 μM , so the indicated concentrations are not possible. Moreover in the experiments where terpenes were pre-dissolved (line 290) and added to autoclaved medium the concentration would have to be even higher to reach 100 or 200 μM . As no concentrations are given, I can not judge if the other 2 terpenes would also have been close or above their solubility in water.

So at best only the results for nerolidol are questionable, at worst growth curves for the other 2 terpenes are also in question.

Response:

The reviewer is correct about the solubility issue for nerolidol. Linalool and cineole have 3.5 g/L and 1.59 g/L water solubility, respectively, that exceed those used in the experiment. While nerolidol reached saturation when mixed in water in our experiment, 10 ml of the unseparated mixture including both dissolved terpene and residual non-dissolved oil was delivered in the plant-microbe systems. It was shown with the sesquiterpene beta-caryophyllene, which is approximately 1000x less soluble in water than nerolidol, that direct injection of the compound into the soil resulted in high bioactivity and rapid diffusion, which was even increased through high moisture. Accordingly, the solid media plates contain an unseparated mixture including dissolved nerolidol and oil suspension in a semi-closed system sealed with parafilm. Similar to the solid media, liquid media used also contains a uniform mixture due to constant shaking.

Hence, we hypothesize that we have retained the described bioavailability of nerolidol with both formulations.

We have clarified this in line 147-149: For nerolidol, the 100 μM was delivered in a solution containing well-homogenized dissolved terpene and undissolved oil as the targeted concentration is beyond the nerolidol water solubility.

Line 149-151: A recent study demonstrated direct delivery and diffusion of an insoluble terpenoid in soils of different moisture levels and demonstrated bioactivity (Chiriboga et al., 2017).

Minor comments would be to include information in the introduction about terpenes that are produced by sorghum, especially in the roots.

Response: Despite some knowledge on terpenes in the aboveground tissues of sorghum, the root metabolism of terpenes is unknown and currently under investigation. We have added specific examples of above-ground terpene metabolism in sorghum and of root accumulating terpenoids from the monocot species switchgrass and maize. This context is now clarified in the introduction in line 85-92.

Line 116: Are the dimensions for one compartment or or all three together?

Response: Explanation added in line 140.

...to rhizoboxes, tri-compartmented acrylic boxes (54 cm x 4 cm x 30 cm) with transparent cover allowing the visualization of root growth...

Line 126: How does the artificial root system look like, is there a reference or a vendor?

Response: Figure S1 shows the installation of the artificial root system. Manufacturer information is added in the material method section in line 154.

Line 194/195: What m/z range was used during detection?

Response: We used 35-250 m/z. This information was added in line 436.

May 24, 2023

Dr. Gregory Bonito
Michigan State University

Re: Spectrum01332-23R1 (Terpenes modulate bacterial and fungal growth and sorghum rhizobiome communities)

Dear Greg/Dr. Bonito:

Thank you for your submitted revision. I am conditionally accepting your revision provided you add/modify a few lines in the Methods section that is more in line with your response to Reviewer #2, which I believe should be in the manuscript directly.

For Lines 146-153 (mid-line), please REPLACE with the following (for text indicated in square brackets "[...]" please replace with appropriate/correct text):

"Four terpenoid treatments, 1,8-cineole 100 μM (water solubility limit = [XX μM] (1.59 g/L)), linalool 100 μM , linalool 200 μM (solubility = [XX μM] (3.5 g/L)), and cis/trans-nerolidol 100 μM (solubility = 63 μM (0.14 g/L)), were used in this study along with a box where only water was added as a control, and a control box with no plants also with only water added. The three terpenoids were diluted with water to obtain the selected concentrations. For nerolidol, the 100 μM was delivered in a solution containing well-homogenized dissolved terpene and undissolved oil as this targeted concentration is beyond the nerolidol water solubility. Ten ml of the unseparated mixture including both dissolved terpene and residual non-dissolved oil was delivered in the plant-microbe systems. Solid media plates contained an unseparated mixture including dissolved nerolidol and oil suspension in a semi-closed system sealed with parafilm while liquid media contained a uniform mixture due to constant shaking. Hence, we hypothesize that the described bioavailability of nerolidol was achieved in both formulations. A recent study demonstrated direct delivery and diffusion of an insoluble terpenoid (the sesquiterpene beta-caryophyllene, with a water solubility $\sim 1000\times$ less than nerolidol) in soils of different moisture levels with demonstrated bioactivity (20)."

Please modify the manuscript along the lines I have recommended. As these revisions are quite minor, I expect that you should be able to turn in the revised paper in less than 30 days, if not sooner. If your manuscript was reviewed, you will find the reviewers' comments below.

When submitting the revised version of your paper, please provide (1) point-by-point responses to the issues raised by the reviewers as file type "Response to Reviewers," not in your cover letter, and (2) a PDF file that indicates the changes from the original submission (by highlighting or underlining the changes) as file type "Marked Up Manuscript - For Review Only". Please use this link to submit your revised manuscript. Detailed instructions on submitting your revised paper are below.

Link Not Available

Sincerely,

Erik Hom

Reviewer comments:

Preparing Revision Guidelines

To submit your modified manuscript, log onto the eJP submission site at <https://spectrum.msubmit.net/cgi-bin/main.plex>. Go to

Author Tasks and click the appropriate manuscript title to begin the revision process. The information that you entered when you first submitted the paper will be displayed. Please update the information as necessary. Here are a few examples of required updates that authors must address:

Please return the manuscript within 60 days; if you cannot complete the modification within this time period, please contact me. If you do not wish to modify the manuscript and prefer to submit it to another journal, please notify me of your decision immediately so that the manuscript may be formally withdrawn from consideration by Microbiology Spectrum.

Response to Editor

Following your suggestion, we have amended the section below, highlighted in the “Marked Up Manuscript- For Review Only”:

Four terpenoid treatments, 1,8-cineole 100 μM (water solubility limit 3.5 g/L [22.7 mM]), linalool 100 μM , linalool 200 μM (water solubility limit 1.6 g/L [10.3 mM]), and cis/trans-nerolidol 100 μM (water solubility limit 14 mg/L [63 μM]), were used in this study along with a box where only water was added as a control, and a control box with no plants also with only water added. The three terpenoids were diluted with water to obtain the selected concentrations. For nerolidol, the 100 μM was delivered in a solution containing well-homogenized dissolved terpene and undissolved oil as the targeted concentration is beyond the nerolidol water solubility. 10 ml of the unseparated mixture including both dissolved terpene and residual non-dissolved oil was delivered in the plant-microbe systems. Solid media plates contain an unseparated mixture including dissolved nerolidol and oil suspension in a semi-closed system sealed with parafilm, while liquid media contained a uniform mixture due to constant shaking. Hence, we hypothesize that the described bioavailability of nerolidol was achieved in both formulations. A recent study demonstrated direct delivery and diffusion of an insoluble terpenoid (the sesquiterpene beta-caryophyllene, with a water solubility $\sim 1000\times$ less than nerolidol) in soils of different moisture levels with demonstrated bioactivity (20).

July 5, 2023

Dr. Gregory Bonito
Michigan State University
Plant Soil and Microbial Sciences
East Lansing, MI

Re: Spectrum01332-23R2 (Terpenes modulate bacterial and fungal growth and sorghum rhizobiome communities)

Dear Greg,

Your manuscript has been accepted, and I am forwarding it to the ASM Journals Department for publication. You will be notified when your proofs are ready to be viewed.

Sincerely,

Erik Hom
Editor, Microbiology Spectrum
